# Psychometric Validation of the Simplified Chinese Version of the Dyspnoea-12 Questionnaire for Patients with Primary Lung Cancer

**DOI:** 10.3390/healthcare13020201

**Published:** 2025-01-20

**Authors:** Xianliang Liu, Bo Peng, Tao Wang, Alex Molassiotis, Janelle Yorke, Liqun Yao, Silin Zheng, Jing-Yu (Benjamin) Tan, Houqiang Huang

**Affiliations:** 1School of Nursing and Health Sciences, Hong Kong Metropolitan University, Homantin, Kowloon, Hong Kong SAR, China; 2Faculty of Health, Charles Darwin University, Brisbane, QLD 4000, Australia; 3School of Nursing, The Hong Kong Polytechnic University, Kowloon, Hong Kong SAR, China; 4The Affiliated Hospital of Southwest Medical University, Luzhou 646000, China; 5College of Arts Humanities and Education, The University of Derby, Derby DE22 1GB, UK; 6School of Nursing, Fujian University of Traditional Chinese Medicine, Fuzhou 350108, China; 7School of Nursing and Midwifery, University of Southern Queensland, Ipswich, QLD 4305, Australia; 8Centre for Health Research, University of Southern Queensland, Springfield, QLD 4300, Australia

**Keywords:** Dyspnoea-12 Questionnaire, psychometric properties, simplified Chinese version, primary lung cancer, validation

## Abstract

**Purpose:** The simplified Chinese version of the Dyspnoea-12 Questionnaire (D-12) has not yet been translated and validated for patients with primary lung cancer. This study aimed to evaluate the psychometric properties of the simplified Chinese version of the D-12 for patients with primary lung cancer. **Methods:** This study analysed the baseline data of a randomised controlled trial that used an inspiratory muscle training intervention for patients with thoracic malignancies. The original English version of the D-12 was translated into simplified Chinese according to standard instrument translation and adaptation procedures. The internal consistency reliability of the D-12 was determined by calculating Cronbach’s alpha coefficients. The convergent validity of the D-12 was evaluated by Spearman’s correlation with the Borg CR-10 Scale, Numerical Rating Scale (NRS), Hospital Anxiety and Depression Scale (HADS), and Saint George’s Respiratory Questionnaire (SGRQ). Blood oxygen level, the 6-minute walk test distance, alcohol use, surgery type, cancer stage, exercise level, and educational background were identified to evaluate their discriminating performance. **Results:** The analysis included 196 participants. The Cronbach’s alpha coefficients for the full D-12 and its physical and emotional function subscales were 0.83, 0.74, and 0.92, respectively. Significantly positive associations were found between the D-12 scores and the Borg CR-10 Scale, the NRS, the HADS, and SGRQ scores (*p* < 0.01). The participants with insomnia (*p* < 0.01) and who did not use alcohol (*p* = 0.019) reported significantly higher D-12 total scores compared with their respective counterparts. The participants at different cancer stages (*p* < 0.01) and those who had undergone different surgeries (*p* = 0.033) reported significantly different D-12 total scores. **Conclusions:** The D-12 simplified Chinese version demonstrated very good psychometric properties and high acceptability in patients with primary lung cancer.

## 1. Introduction

Dyspnoea (breathlessness) is defined as “a subjective experience of breathing discomfort that consists of qualitatively distinct sensations that vary in intensity” (p. 436) [1]. Dyspnoea is a highly prevalent and distressing symptom in primary lung cancer, with an incidence rate as high as 90% in advanced lung cancer [2,3]. Previous studies have reported that dyspnoea symptoms more frequently occur in lung cancer patients than in patients with other types of cancer [4,5]. The dyspnoea symptoms of patients with primary lung cancer tend to be more frequent and severe during different physical activities, such as walking, dressing, and working [6,7]. Furthermore, dyspnoea progressively interferes with physical and psychosocial functioning and has profound negative effects on multiple domains of quality of life (QoL) in patients with lung cancer [8]. Despite increasing recognition of the significance of dyspnoea, it remains poorly undertreated in clinical settings as an undetected symptom [9,10].

The commonly used instruments for dyspnoea assessment are the Borg Scale [11], the Visual Analogue Scale (VAS) [12], the Numerical Rating Scale (NRS) [13], the Medical Research Council (MRC) Dyspnoea Scale [14], the Cancer Dyspnoea Scale (CDS) [15,16], the Chronic Respiratory Questionnaire (CRQ) [17] and Saint George’s Respiratory Questionnaire (SGRQ) [18]. However, the Borg Scale, the VAS, and the NRS are unidimensional measurements that focus exclusively on the severity of the sensation [11,12]. These unidimensional assessment tools cannot capture sufficient information on multiple important dimensions of dyspnoea, such as the sensory and affective qualities of dyspnoea and its functional impacts onpatients’ QoL [12]. The CDS only measures the psychological properties of anxiety, and other affective components of dyspnoea are missing from the scale [15]. Moreover, the MRC Dyspnoea Scale, SGRQ, and CRQ indirectly assess dyspnoea and do not or only partly capture the sensory experience and the affective components of the symptoms [19,20]. Therefore, standardised multidimensional instruments that directly assess and capture different aspects of dyspnoea in patients with primary lung cancer are needed.

The Dyspnoea-12 Questionnaire (D-12) is a 12-item multidimensional, convenient, and self-reported instrument that was originally developed and validated for patients with chronic obstructive pulmonary disease (COPD) and heart failure by Yorke and colleagues [21]. The D-12 includes both sensory experience (“physical functions”, seven items) and affective (“emotional functions”, five items) components of dyspnoea symptoms [21]. Compared to other multidimensional scales of dyspnoea, such as the Baseline/Transitional Dyspnoea Index and the Multidimensional Dyspnoea Profile, the D-12 is designed to be a quick and easy-to-complete scale for assessing dyspnoea, making it particularly useful in busy clinical settings [22]. The original D-12 English version has subsequently demonstrated satisfactory validity and reliability for the assessment of dyspnoea (breathlessness) in different health conditions, such as asthma [23], interstitial lung disease [24], pulmonary hypertension [25] and lung cancer [26]. In addition to the original English version, Arabic [27], French [28], Korean [29], Japanese [30], Italian [31], Portuguese [32], Norwegian [33], Swedish [34] and traditional Chinese [35] versions of the D-12 are available. The traditional Chinese version of the D-12 demonstrated very good validity and reliability in measuring dyspnoea among patients with COPD [35], but the simplified Chinese version of the D-12 has not yet been translated and validated for patients with primary lung cancer. Simplified Chinese characters are mainly used in Mainland China, and traditional Chinese mainly used in Hong Kong, Macau and Taiwan. Traditional Chinese shows a higher level of complexity compared with the simplified Chinese characters [36]. Moreover, due to the complexity of culture, healthcare systems, and other social, environmental factors among the two groups of Chinese users, the traditional Chinese version of the D-12 cannot be used directly by populations living in Mainland China. Taken together, this study aimed to translate and evaluate the acceptability, validity, and reliability of the simplified Chinese version of the D-12 to measure dyspnoea (breathlessness) in patients with primary lung cancer.

## 2. Methods

This adaptation and validation study was a secondary analysis of baseline data obtained from a randomised controlled trial (RCT, NCT03834116) that assessed the effectiveness of an inspiratory muscle training intervention for dyspnoea in patients with thoracic malignancies. The sample size estimation was determined based on the primary outcome, mBorg score, and a 25% attrition rate observed in the pilot study [37]; the required sample size was 196, with 98 in each group. The data were collected between 2020 and 2021. Ethical approvals for the RCT were obtained from the Clinical Trial Research Ethics Committee at the Affiliated Hospital of Southwest Medical University (KY2018002), the Human Subjects Ethics Sub-Committee at The Hong Kong Polytechnic University (HSEARS20180509003), and the Human Research Ethics Committee at Charles Darwin University (H19013). All of the participants provided written informed consent before participating in the RCT.

### 2.1. Overview of the Trial

The study site of the two-arm, non-blinded RCT was the Affiliated Hospital of Southwest Medical University, Luzhou, Sichuan Province, China. Adults with a diagnosis of primary lung cancer were included in this study. The Borg CR-10 Scale, the D-12, the NRS (perceived severity of breathlessness), the Hospital Anxiety and Depression Scale (HADS), and SGRQ were used to measure the effectiveness of the intervention. The current adaptation and validation study only analysed baseline data from all the participants who completed the baseline assessment of this RCT.

### 2.2. Translation Process

The original author granted permission to translate the D-12 from English into simplified Chinese and validate the D-12 in the Chinese population. To ensure meaning accuracy and cultural acceptability, the original English version of the D-12 was translated into simplified Chinese in accordance with the standard instrument translation, adaptation, and validation procedures [38,39]. Two bilingual translators (XL-L and JY-T) in both English and simplified Chinese independently translated the original English version of the D-12 into simplified Chinese. These two translators (XL-L and JY-T) were senior researchers in cancer symptom management and familiar with the terminology related to dyspnoea. Any ambiguities and discrepancies were discussed and resolved with the research team, which included two professors in nursing, two nurse managers, one oncology physician, and two oncology nurses. Then, a blind backward translation of the instrument was performed by another two bilingual oncology nurses who worked at the study site. The research team analysed the cultural appropriateness and measurement equivalence of the D-12 simplified Chinese version, and then the words and phrases that diverged from the original D-12 English version were compared and discussed with the translators. Minor modifications of the D-12 simplified Chinese version were incorporated to achieve a linguistically and culturally equivalent meaning between the original English version and the forward simplified Chinese version. To test the clarity and readability of the D-12 simplified Chinese version, pilot testing with ten patients with primary lung cancer with different education levels at the study site was conducted using a convenient sampling method. Patients were asked to fill out the prefinal D-12 simplified Chinese version and highlighted if any sentences or words in each D-12 item were ambiguous, discrepant and sensitive. All completed D-12 questionnaires in paper form from this step were re-evaluated by the research team. The patients indicated that there was no need to modify the D-12 simplified Chinese version, as all items were clear, understandable, and culturally appropriate.

### 2.3. Study Instruments

The following instruments used for the RCT were selected to evaluate the acceptability, validity, and reliability of the D-12 simplified Chinese version for patients with primary lung cancer.

#### 2.3.1. Demographic and Clinical Characteristics Data

The participants’ demographic and clinical characteristics were collected using a self-developed data collection form, including age, educational background, marital status, occupation, body mass index (BMI), cancer stage, surgery type, and current or previous health conditions related to dyspnoea symptoms, such as blood oxygen level.

#### 2.3.2. Dyspnoea-12 Questionnaire (D-12)

The D-12 uses 12 items to obtain an overview of the severity of breathlessness symptoms and their impact on a patient’s physical and emotional well-being [24]. The D-12 total score ranges from 0 to 36, with a high score indicating worse dyspnoea [24]. The D-12 includes a physical component (items 1 to 7) and an emotional component (items 8 to 12) [26].

#### 2.3.3. Borg CR-10 Scale

The Borg CR-10 Scale is the most widely used breathlessness scale. The 0–10 categorical scale has ratio-level properties [11]. Participants are asked to select words or a number that most appropriately describes their sensation of breathlessness, from 0 (“nothing at all”) to 10 (“maximal”) [40]. The Borg CR-10 scale was previously used to evaluate dyspnoea and breathlessness in patients with COPD [20] and lung cancer [41].

#### 2.3.4. Numerical Rating Scale (NRS)

The perceived severity of breathlessness (including the “worst” breathlessness over the past 24 h and current breathlessness), also measured by a 0–10 categorical scale, was anchored as follows: 0 = “no breathlessness at all” and 10 = “worst imaginable breathlessness” [42]. The NRS is a commonly used tool because it provides easier access to subjects’ experiences with dyspnoea in general assessment settings [13].

#### 2.3.5. Hospital Anxiety and Depression Scale (HADS)

The HADS is a 14-item well-established scale that measures the frequency of anxiety and depression over the previous four weeks [43]. All the items in the HADS are rated on a scale between 0 and 3, and a higher HADS score indicates more severe depression or anxiety symptoms [43]. The HADS is commonly used and has been validated in cancer patients [44,45].

#### 2.3.6. Saint George’s Respiratory Questionnaire (SGRQ)

The SGRQ consists of 76 items, including “symptoms”, “activity”, “impact”, and “total” scores [46]. Each item in the SGRQ has a specific weight, which is combined to calculate the subscale and total scores [18]. The SGRQ total score ranges from 0 to 100, with a high score indicating more limitations. The simplified Chinese version of SGRQ is a reliable, valid, and responsive tool for measuring the QoL of Chinese patients with lung diseases [47].

### 2.4. Acceptability and Psychometric Assessment

#### 2.4.1. Acceptability and Floor and Ceiling Effects

The acceptability of the D-12 was evaluated by the proportion of missing data of all items in the D-12. The proportion of the participants who reported the highest and lowest total scores was used to identify the ceiling and floor effects, respectively. The presence of ceiling and floor effects was determined by 15% of the participants with the highest and lowest total scores [48].

#### 2.4.2. Reliability

Reliability in the D-12 was demonstrated by internal consistency reliability, which reflects the extent to which the items within a questionnaire are interrelated [49]. The internal consistency reliability of the D-12 total and subscale scores was evaluated through the calculation of the corrected item–total correlations and the Cronbach’s alpha coefficients of all the D-12 items as well as those in the physical and emotional function subscales. The association between one single D-12 item score and the D-12 total score excluding that item was calculated by the corrected item–total correlations of all the D-12 items. If the Cronbach’s alpha and Spearman–Brown coefficients were a minimum of 0.70, this indicated a satisfactory internal consistency and reflected a satisfactory relationship between the D-12 items [50]. An item was considered satisfactory if the corrected item-to-total correlation coefficient was 0.30 or more [51].

#### 2.4.3. Validity

Convergent validity reflects the extent to which several scales capture a common concept or construct [52], and it was calculated to identify the validity of the D-12. Correlations between the D-12 (total and subscales) scores and the Borg CR-10 Scale, the NRS (perceived severity of breathlessness), the HADS (total and subscales), and SGRQ (total and subscales) scores were calculated to determine the convergent validity of the D-12. A moderate or strong correlation was determined by a correlation coefficient of more than 0.30 or more than 0.50, respectively [53].

#### 2.4.4. Discriminating Performance and Invariance

Multigroup analysis was used to evaluate the invariance of the D-12. Blood oxygen level and the 6-minute walk test distance (6-MWTD) were selected for multigroup analysis, as the literature has indicated that blood oxygen level [54] and the 6-MWTD [55] were correlated with dyspnoea symptoms. The discriminating performance of the D-12 was evaluated through subgroup analysis of the participants’ use or non-use of alcohol and the presence or not of insomnia, as well as those who had received different surgeries, were at different cancer stages, and had different exercise levels and educational backgrounds [56,57,58,59].

### 2.5. Statistical Analysis

All data analyses were performed using IBM SPSS Statistics version 28.0 for Windows. Descriptive analyses were performed for all included variables. A normality test determined that most of the D-12, Borg CR-10 Scale, NRS (perceived severity of breathlessness), HADS, and SGRQ scores violated the assumption of normal distribution in the current study’s population; therefore, Spearman’s correlation was selected to determine the correlations between the D-12 (total and subscales) scores and the Borg CR-10 Scale, the NRS (perceived severity of breathlessness), the HADS (total and subscales), and SGRQ (total and subscales) scores. The Mann–Whitney U test or independent *t*-test and the Kruskal–Wallis H test or a one-way analysis of variance (ANOVA) were used to evaluate the group differences in the D-12 total scores according to the results of the normality tests. A two-tailed *p* < 0.05 was considered statistically significant.

## 3. Results

### 3.1. Sample Characteristics

One hundred and ninety-six patients were recruited from the outpatient department at the study site. All participants were diagnosed with lung cancer, with different cancer stages ranging from stage I to IV. The majority of them completed the surgical procedure. The mean age was 58.8 years (range = 20 to 82; SD = 9.9), the mean blood oxygen level (%) was 97.6 (SD = 1.4), and the mean BMI was 23.1 (SD = 3.1) in this study sample. The participants’ demographic and clinical information is presented in Table 1.

### 3.2. Acceptability and Floor and Ceiling Effects

The proportion of missing values ranged from 0.5% to 1.5%, and missing data were identified in all the D-12 items. Four participants did not respond to at least one of the items, which led to a missing value rate of 2.0% at a scale level, as shown in Table 2. The floor and ceiling effects were 0% at a scale level, as there were no participants with the lowest (0) and highest (36) D-12 total scores. However, 10 of the 12 items as well as the emotional function subscale showed a very large floor effect, as shown in Table 2.

### 3.3. Reliability of the D-12

The mean total score of the D-12 was 7.7 (SD = 3.9; range: 2 to 27; see Table 3). The D-12 total and subscale scores presented excellent or satisfactory internal consistency, with a Cronbach’s alpha coefficient of 0.83 (full D-12), 0.74 (physical function subscale), and 0.92 (emotional function subscale). The Cronbach’s alpha values were more than 0.80 after deleting each of the D-12 items. Excellent item-to-total correlations were determined for 11 items (range: 0.40 to 0.66), and one item-to-total correlation was satisfactory (0.30; see Table 3).

### 3.4. Convergent Validity of the D-12

All 196 participants were required to complete several questionnaires during the baseline assessments, including the Borg CR-10 Scale (n = 195), the D-12 (n = 192), the NRS (which measures perceived severity of breathlessness, n = 195), the HADS (n = 191), and the SGRQ (n = 196). Out of the 196 participants, only a few (between 0 and 5) were without a total score for these questionnaires because these participants did not complete all questions within the questionnaires. The mean total scores for the Borg CR-10 Scale, HADS, and SGRQ were 4.2 (n = 195), 4.9 (n = 191), and 28.2 (n = 196), respectively, as shown in Table 4. Furthermore, the mean subscale scores for the HADS were 3.1 for anxiety (n = 195) and 1.8 for depression (n = 191). The majority of participants did not have anxiety or depression, and only 11 participants experienced minor anxiety, while 4 participants had moderate anxiety [43]. Similarly, four participants had minor depression, and another four participants experienced moderate depression [43]. The SGRQ total and subscale scores were relatively low, indicating fewer limitations [47].

The D-12 total and physical function subscale scores were significantly and positively associated with the Borg CR-10 Scale, the NRS (“worst” breathlessness over the past 24 h and current breathlessness), the HADS (total and subscales), and SGRQ (total and subscales) scores (*p* < 0.01), and the correlation was moderate to strong, with correlation coefficients ranging from 0.30 to 0.62. Similarly, the D-12 emotional function subscale was significantly positively correlated with the HADS (total and subscales) and SGRQ (total and subscales) scores (*p* < 0.01), and the correlation coefficients ranged from 0.28 to 0.52. More information is shown in Table 5.

### 3.5. Discriminating Performance and Invariance of the D-12

The D-12 total scores and blood oxygen level demonstrated a significantly negative but weak correlation (*r* = −0.25, *p* ˂ 0.01, n = 192). The D-12 total scores and the 6-MWTD showed a significantly negative and moderate correlation (*r* = −0.51, *p* ˂ 0.01, n = 191). The mean 6-MWTD was 413.7 (SD 59.4) meters. The patients with insomnia had significantly higher D-12 total scores (indicating a higher level of dyspnoea) than those without insomnia (*z* = −4.47, *p* < 0.01, n = 192), and similar results were reported by the participants who did not use alcohol (*z* = −2.34, *p* = 0.019, n = 192; see Table 5). The participants at different cancer stages reported significant differences in the D-12 total scores (*p* < 0.01, n = 180), as did the participants who had received different surgery types (*p* = 0.033, n = 166; see Table 6). There was no difference in the D-12 total scores between the patients with different exercise durations (*p* > 0.05, n = 191), and patients with different educational backgrounds (*p* > 0.05, n = 192) showed similar D-12 total scores.

## 4. Discussion

This study demonstrated the very good psychometric properties and high acceptability of the D-12 simplified Chinese version in patients with primary lung cancer. The D-12 translation procedure in this study was straightforward, following standard instrument translation processes [38,39], and the pilot testing with ten patients indicated that all 12 items were clear, understandable, and culturally applicable. The internal consistency reliability of the D-12 simplified Chinese version was high, with a Cronbach’s alpha value of 0.83 for the full D-12. These results demonstrated satisfactory corrected item–total coefficients (>0.3) for all the D-12 items and indicated that the simplified Chinese version of the D-12 was internally consistent. The D-12 simplified Chinese version also showed excellent convergent validity in this study. The participants experienced mild to moderate dyspnoea symptoms, with mild impacts on physical functions (the subscale score was 7.0) and almost no impacts on emotional functions (the mean subscale score was 0.7). The majority of participants did not have anxiety or depression according to the HADS, and the SGRQ total and subscale scores were relatively low, indicating fewer limitations. These might also explain the considerable floor effects identified in the emotional functions’ subscale.

The validity and reliability of the simplified Chinese version of the D-12 in this study appeared to be quite similar to the excellent psychometric properties reported for the English [26] and Japanese [30] versions of the D-12 for patients with lung cancer. The original English version of the D-12 was validated by 101 patients with lung cancer, and it demonstrated adequate convergent validity and excellent reliability, with a Cronbach’s alpha value of 0.95 for the D-12 total scores [26]. Similar to the original English version, the D-12 Japanese version also confirmed excellent validity and reliability for 113 patients with lung cancer, with a Cronbach’s alpha value of 0.97 for the D-12 total scores. In comparison, the D-12 simplified Chinese version reported a lower Cronbach’s alpha value (0.83) in this study. This may be explained by the finding that 80 to 90% of the patients reported a score of 0 in all of the D-12 emotional function subscale items (items 8 to 12) and a Cronbach’s alpha value of 0.74 [60]. The D-12 traditional Chinese version validation study also found a similar issue in that 26% of the participants attained the lowest D-12 total score (0), and the authors explained that Chinese people tend to underreport symptoms and psychological distress [35]. Furthermore, unlike earlier validation studies conducted in Arabic, French, and traditional Chinese, which focused on different cultural and healthcare contexts, this study fills a critical gap by focusing on the unique sociocultural contexts of Mainland China. The differences in healthcare access and patient engagement between Mainland China and regions like Hong Kong—where the Traditional Chinese version was validated—highlight the need for localised assessment tools. These findings highlight the importance of culturally relevant instruments that consider regional linguistic and healthcare complexities, thereby improving the D-12’s clinical utility for assessing dyspnoea in diverse populations.

The convergent validity of the simplified Chinese version of the D-12 was excellent. As hypothesised, the D-12 total scores moderately or strongly correlated with the Borg CR-10 Scale scores, the NRS scores, the HADS scores, and the SGRQ scores. The D-12 includes seven physical function items and five emotional function items, which measure very similar or the same concepts as the Borg CR-10 Scale [11], the NRS (perceived severity of breathlessness) [13], the HADS [43], and the SGRQ [46]. For example, both the HADS and D-12 contain emotional content for measuring psychological distress. The HADS and D-12 were used to explore the association between them [26], and the HADS is commonly utilized and has been validated in cancer patients [44,45]. Positive associations between the D-12 and the NRS, the HADS, and SGRQ were also reported by other similar studies [26,28,30,35]. Furthermore, other studies reported that dyspnoea significantly affects the QoL of patients with advanced cancer [61,62], which is consistent with the findings of the present study.

The simplified Chinese version of the D-12 demonstrated excellent discriminating performance via subgroup analysis, with well-known influencing factors for dyspnoea. The study results showed that the D-12 total scores were significantly negatively associated with blood oxygen level and the 6-MWTD. This is consistent with previous study findings that dyspnoea leads to lower blood oxygen levels [54] and decreased exercise capacity [55]. Moreover, the participants who had experienced insomnia reported higher levels of dyspnoea with more physical and emotional impacts than those without insomnia. This is consistent with prior findings that dyspnoea severity contributes to higher sleep disturbances and bad subjective sleep quality [56]. The D-12 simplified Chinese version was sensitive to the dyspnoea measurements as captured by the differences in dyspnoea symptoms in patients with different blood oxygen levels, exercise capacities, and sleep quality in this study. A multinational, prospective, and longitudinal study reported that lung metastases (end-stage cancers) were independent predictors of worse dyspnoea [58]. In addition, types of surgery were treatment-related causes of dyspnoea [57]. The participants at different cancer stages and those who had received different surgeries reported significantly different D-12 total scores in this study, which indicated that the D-12 simplified Chinese version accurately captured the differences in dyspnoea in participants at different disease stages and who had received different surgeries. Furthermore, patients with low alcohol consumption reported a lower incidence of respiratory symptoms compared with non-drinkers [59]. This was also accurately and sensitively captured by the D-12, as participants who used alcohol reported lower levels of dyspnoea with fewer physical and emotional impacts than non-drinkers.

### Study Limitations

Although the results of this study are consistent with those in previous studies, several limitations in the present study should be considered. Test–retest reliability is an important psychometric property for the validation of an instrument, but it was not assessed as this study was a secondary analysis of baseline data from an RCT on the poor stability of dyspnoea symptoms. Future research may benefit from assessing the test–retest reliability to further validate the D-12. A considerable floor effect was observed for all items in the emotional function subscale of the simplified Chinese version of the D-12. Other validation studies involving D-12 in Japan [30] and Hong Kong [35] identified similar floor effects in the emotional component. Given the similar demographic features of these Asian regions, the observed floor effect in the emotional component may be attributed to Chinese or Asian cultural norms, characterised by a common tendency to practice emotional restraint and avoid expressing negative emotions openly [63]. Cultural tendencies towards emotional restraint may lead patients to underreport their emotional issues. Consequently, it may be crucial for healthcare professionals to employ culturally sensitive approaches that encourage emotional expression when utilising the D-12 with this population [64]. However, the majority of participants did not experience anxiety or depression, which may also significantly contribute to the floor effects observed in the emotional subscale of the D-12. Therefore, the observed considerable floor effect in the D-12 emotional function subscale likely reflects the true perceptions of affect and emotions in this population, as evidenced by scores across all patient-reported measures of distress, such as the HADS and SGRQ impacts, which were negligible. In addition, this study was limited to individuals with primary lung cancer, and further testing in other Chinese populations and settings is necessary. Furthermore, more advanced psychometric testing, such as Confirmatory Factor Analysis (CFA), is unsuitable for this study. A two-factor CFA requires a sample size of 450 respondents with three or four items per scale and factor loadings of at least 0.5 [65]. In this study, the D-12 measure comprises two distinct factors to be assessed—a physical component factor (items 1 to 7) and an emotional component factor (items 8 to 12)—both with factor loadings of more than 0.5. Therefore, the sample size of 196 participants in this study may not provide adequate power for a reliable CFA.

## 5. Conclusions

The simplified Chinese version of the D-12 was demonstrated to be a sufficiently valid and reliable instrument for assessing dyspnoea and its impacts on patients with primary lung cancer. The simplified Chinese version of the D-12 will provide healthcare professionals and researchers with an easy instrument for screening and assessing dyspnoea and its impacts on Chinese-speaking patients with primary lung cancer.

## Figures and Tables

**Table 1 healthcare-13-00201-t001:** Demographic and clinical information.

Demographic and Clinical Information		Number (%)
Educational background (n = 196)	No formal education/uneducated	20 (10.2)
	Primary school	50 (25.5)
	Secondary school	71 (36.2)
	High school or vocational school	21 (10.7)
	College diploma	18 (9.2)
	University degree or above	16 (8.2)
Marital status (n = 196)	Single	2 (1.0)
	Married	194 (99.0)
Gender (n = 196)	Male	91 (46.4)
	Female	105 (53.6)
Occupation (n = 194)	Professional and technical personnel	16 (8.2)
	Manual worker	49 (25.0)
	Clerical or administrative worker	10 (5.1)
	No longer working *	77 (39.3)
	Other	42 (21.4)
Household income (RMB/month) (n = 178)	<3000	106 (54.1)
	3000–6000	61 (31.1)
	6001–10,000	10 (5.1)
	>10,000	1 (0.5)
Source of healthcare insurance (n = 195)	NCMS	88 (44.9)
	URBMI and UEBMI	104 (53.1)
	Self-paid	3 (1.5)
BMI (n = 194)	Underweight (<18.5)	12 (6.1)
	Normal or healthy weight (18.5–22.9)	85 (43.4)
	Overweight (23–24.9)	46 (23.5)
	Obese (≥25)	53 (27)
Cancer stage (n = 183)	I	124 (63.3)
	IIA	8 (4.1)
	IIB	10 (5.1)
	IIIA	15 (7.7)
	IIIB	8 (4.1)
	IIIC	1 (0.5)
	IV	17 (8.7)
Surgery types (n = 169)	Left upper lobectomy	46 (23.5)
	Left lower lobectomy	12 (6.1)
	Right upper lobectomy	51 (26.0)
	Right lower lobectomy	33 (16.8)
	Microwave ablation	1 (0.5)
	Other	26 (13.3)
Hypertension (n = 195)	Yes	45 (23.0)
	No	150 (76.5)
Exercise (Hour/week) (n = 195)	0–2	157 (80.1)
	3–4	25 (12.8)
	5–6	9 (4.6)
	>6	4 (2.0)
Smoking (n = 195)	Never smoked	129 (65.8)
	Current smoking	3 (1.5)
	Previous smoking	63 (32.1)
Alcohol consumption (n = 196)	Yes	71 (36.2)
	No	125 (63.8)
Insomnia (n = 196)	Yes	72 (36.7)
	No	124 (63.3)
Diabetes (n = 196)	Yes	15 (7.7)
	No	181 (93.2)
Asthma (n = 196)	Yes	4 (2.0)
	No	192 (98.0)
Pneumonectasis (n = 196)	Yes	8 (4.1)
	No	188 (95.9)
Pulmonary tuberculosis (n = 196)	Yes	6 (3.1)
	No	190 (96.9)
Heart diseases (n = 194)	Yes	12 (6.1)
	No	182 (92.9)

Note: BMI, body mass index; NCMS, the rural new cooperative medical scheme; UEBMI, the urban employee-based basic medical insurance; URBMI, urban resident-based basic medical insurance scheme. * includes housewives and unemployed and retired people.

**Table 2 healthcare-13-00201-t002:** Item analysis (n = 196).

	No. of Subjects Who Responded to the Item	No. of Subjects Who Did Not Respond to the Item	Missing Value (%)	Floor Effect (%)	Ceiling Effect (%)
D-12 item 1	195	1	0.5%	17.4%	1.0%
D-12 item 2	195	1	0.5%	25.5%	0.5%
D-12 item 3	195	1	0.5%	8.2%	2.6%
D-12 item 4	195	1	0.5%	12.8%	2.0%
D-12 item 5	195	1	0.5%	25.0%	0.5%
D-12 item 6	195	1	0.5%	17.3%	0.5%
D-12 item 7	195	1	0.5%	77.0%	0.5%
D-12 item 8	195	1	0.5%	91.8%	0%
D-12 item 9	194	2	1.0%	91.3%	0%
D-12 item 10	193	3	1.5%	91.3%	0%
D-12 item 11	195	1	0.5%	82.7%	0%
D-12 item 12	195	1	0.5%	81.6%	0%
D-12 Physical	195	1	0.5%	0%	0.5%
D-12 Emotional	192	4	2.0%	79.1%	0%
	No. of subjects who responded to all the items	No. of subjects who did not respond to one or more of the items			
D-12 total	192	4	2.0%	0%	0%

Note: D-12: Dyspnoea-12 Questionnaire; D-12 Physical: item 1–7; D-12 Emotional: item 8–12. Each question is scored from 0 to 3, with 3 indicating greater dyspnoea severity.

**Table 3 healthcare-13-00201-t003:** Reliability of the D-12.

Question Number and Content		n	Mean Score	SD	Corrected Item–Total Correlation	Cronbach’s Alpha if Item Deleted
Item 1	My breath does not go in all the way	195	1.1	0.7	0.47	0.82
Item 2	My breathing requires more work	195	0.9	0.7	0.47	0.82
Item 3	I feel short of breath	195	1.4	0.7	0.30	0.83
Item 4	I have difficulty catching my breath	195	1.3	0.7	0.40	0.83
Item 5	I cannot get enough air	195	1.0	0.7	0.49	0.81
Item 6	My breathing is uncomfortable	195	1.0	0.6	0.45	0.82
Item 7	My breathing is exhausting	195	0.3	0.5	0.65	0.80
Item 8	My breathing makes me feel depressed	195	0.1	0.3	0.57	0.82
Item 9	My breathing makes me feel miserable	194	0.1	0.3	0.64	0.81
Item 10	My breathing is distressing	193	0.1	0.3	0.66	0.81
Item 11	My breathing makes me agitated	195	0.2	0.4	0.60	0.81
Item 12	My breathing is irritating	195	0.2	0.5	0.63	0.81
D-12 Physical		195	7.0	2.8		
D-12 Emotional		192	0.7	1.6		
D-12 Total score		192	7.7	3.9		

Note: D-12: Dyspnoea-12 Questionnaire; SD, standard deviation; D-12 Physical: item 1–7; D-12 Emotional: item 8–12. Each question is scored from 0 to 3, with 3 indicating greater dyspnoea severity.

**Table 4 healthcare-13-00201-t004:** Borg, NRS, HADS and SGRQ scores.

Questionnaire/Subscale	n	Mean Score	SD
Borg score	195	4.2	1.5
NRS1	195	4.5	2.1
NRS2	195	2.5	1.3
SGRQ Symptom	196	24.0	14.4
SGRQ Activity	196	44.6	16.0
SGRQ Impact	196	19.9	13.6
SGRQ Total	196	28.2	13.1
HADS-A	195	3.1	2.7
HADS-D	191	1.8	2.5
HADS total	191	4.9	4.8

Note: NRS: numerical rating scale; NRS1: ‘worst’ over the past 24 h; NRS2: current dyspnoea: SGRQ: St George’s Respiratory Questionnaire; HADS: Hospital Anxiety and Depression Scale; A: anxiety; D: depression.

**Table 5 healthcare-13-00201-t005:** Convergent validity of the D-12.

	D-12 Total Score	D-12 Physical	D-12 Emotional	Borg	NRS1	NRS2	SGRQ Symptom	SGRQ Activity	SGRQ Impact	SGRQ Total	HADS-A	HADS-D	HADS Total
D-12 total score	1.000												
D-12 Physical	0.955 **	1.000											
D-12 Emotional	0.554 **	0.340 **	1.000										
Borg	0.456 **	0.533 **	0.075	1.000									
NRS1	0.457 **	0.547 **	−0.023	0.726 **	1.000								
NRS2	0.434 **	0.505 **	0.100	0.703 **	0.827 **	1.000							
SGRQ Symptom	0.352 **	0.422 **	−0.003	0.407 **	0.571 **	0.437 **	1.000						
SGRQ Activity	0.584 **	0.584 **	0.308 **	0.487 **	0.502 **	0.530 **	0.480 **	1.000					
SGRQ Impact	0.553 **	0.573 **	0.302 **	0.511 **	0.547 **	0.583 **	0.577 **	0.690 **	1.000				
SGRQ Total	0.599 **	0.621 **	0.283 **	0.542 **	0.606 **	0.607 **	0.713 **	0.841 **	0.940 **	1.000			
HADS-A	0.415 **	0.343 **	0.443 **	0.242 **	0.233 **	0.321 **	0.205 **	0.297 **	0.460 **	0.421 **	1.000		
HADS-D	0.409 **	0.301 **	0.502 **	−0.064	−0.050	0.096	−0.084	0.188 **	0.279 **	0.211 **	0.517 **	1.000	
HADS total	0.461 **	0.361 **	0.524 **	0.128	0.131	0.262 **	0.100	0.287 **	0.442 **	0.381 **	0.908 **	0.810 **	1.000

Note: D-12: Dyspnoea-12 Questionnaire; NRS: numerical rating scale; NRS1: ‘worst’ over the past 24 h; NRS2: current dyspnoea: SGRQ: St George’s Respiratory Questionnaire; HADS: Hospital Anxiety and Depression Scale; A: anxiety; D: depression; ** Correlation is significant at the 0.01 level (two-tailed).

**Table 6 healthcare-13-00201-t006:** Difference in D-12 total score between different groups.

Demographic and Clinical Group		D-12 Total Score Mean Rank	D-12 Total Score Mean, SD	Z/Kruskal–Wallis H	*p*-Value
Insomnia	Yes (n = 70)	121.4	9.4 (4.7)	−4.47	<0.01
	No (n = 122)	82.2	6.6 (2.8)		
Use of alcohol	Yes (n = 69)	84.1	6.5 (2.4)	−2.34	0.019
	No (n = 123)	103.5	8.3 (4.4)		
Cancer stage	I (n = 121)	85.2	7.1 (2.9)	22.99	<0.01
	IIA (n = 8)	50.5	5.3 (2.7)		
	IIIB (n = 10)	94.3	7.4 (2.6)		
	IIIA (n = 15)	141.0	12.1 (5.5)		
	IIIB (n = 8)	91.8	7.4 (2.9)		
	IIIC (n = 1)	171.5	-		
	IV (n = 17)	95.1	9.3 (6.7)		
Surgery type	Left upper lobectomy (n = 46)	90.7	7.6 (2.8)	13.16	0.033
	Left lower lobectomy (n = 12)	90.8	7.4 (2.5)		
	Right upper lobectomy (n = 50)	74.7	6.6 (2.8)		
	Right lower lobectomy (n = 32)	99.3	9.1 (5.1)		
	Microwave ablation (n = 1)	5.0	-		
	Other (n = 25)	67.3	6.6 (3.5)		

Note: D-12: Dyspnoea-12 Questionnaire; SD, standard deviation.

## Data Availability

The data that support the findings of this study are available on request from the corresponding author. The data are not publicly available due to privacy or ethical restrictions.

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
