# Peer review of "Psychometric Validation of the Simplified Chinese Version of the Dyspnoea-12 Questionnaire for Patients with Primary Lung Cancer"

_healthcare, 2025, doi:10.3390/healthcare13020201_

Round 1
Reviewer 1 Report
Comments and Suggestions for Authors
Good presentation. Here are some questions -
1. Did you look at PFTs for these patients and D-12 association with degree of FEV1 or FEV1/FVC or lung dolumes or diffusion capacity.
2. Can you give more information on D-12 scores and 6MWTD information. What is the average length walked for these patients according to their severity of D12 scores.
3. Can you elaborate on the trial design. a) Did all 196 patients complete all the questionnaires - D12, Borg, HADS, NRS, SGRQ. b) When was these information obtained - on initial diagnosis of cancer, before or after surgery etc. c) Were there any hospitalizations for these patients? d) was there any change in these scores before and after surgery
4. Did you look into association with mortality and D12 scores?
5. Did you look into inhalers and d12 scores? Were these patients on inhalers, if yes how many and was their an association?
6. How did the D12 scores compare with other co-existing diagnosis like Pulmonary hypertension, asthma, ILD, heart failure etc.
Author Response
|
Reviewers |
Authors’ Responses |
|
1. Did you look at PFTs for these patients and D-12 association with degree of FEV1 or FEV1/FVC or lung dolumes or diffusion capacity.
|
Thank you for this suggestion. Since this is a secondary analysis of baseline data obtained from a randomized controlled trial, we did not collect information such as pulmonary function tests (PFTs). Therefore, conducting an association analysis using these PFTs is not feasible. |
|
2. Can you give more information on D-12 scores and 6MWTD information. What is the average length walked for these patients according to their severity of D12 scores.
|
Thank you for this comment. We have inserted the following sentence in the “Discriminating Performance and Invariance of the D-12” section to address this point. Page 10: “The D-12 total scores and the 6-MWTD showed a significantly negative and moderate correlation (r = -0.51, p ˂ 0.01, n = 191). The mean 6-MWTD was 413.7 (SD 59.4) meters.” |
|
3. Can you elaborate on the trial design. a) Did all 196 patients complete all the questionnaires - D12, Borg, HADS, NRS, SGRQ. b) When was these information obtained - on initial diagnosis of cancer, before or after surgery etc. c) Were there any hospitalizations for these patients? d) was there any change in these scores before and after surgery
|
Thank you for these comments. To address these points, we have inserted the following sentences in the “results” section. We described that this adaptation and validation study was a secondary analysis of baseline data obtained from a randomised controlled trial (RCT, NCT03834116) that assessed the effectiveness of an inspiratory muscle training (IMT) intervention for dyspnoea in patients with thoracic malignancies. Therefore, we cannot compare the changes that occurred after the surgery. Page 5: “One hundred and ninety-six patients were recruited from the outpatient department at the study site. All participants were diagnosed with lung cancer, with different cancer stages ranging from stage I to IV. The majority of them completed the surgical procedure. The mean age was 58.8 years (range = 20 to 82; SD = 9.9), the mean blood oxygen level (%) was 97.6 (SD = 1.4), and the mean BMI was 23.1 (SD = 3.1) in this study sample.” Page 8-9: “All 196 participants were required to complete several questionnaires during the baseline assessments, including the Borg CR-10 Scale (n=195), the D-12 (n=192), the NRS (which measures perceived severity of breathlessness, n=195), the HADS (n=191), and the SGRQ (n=196). Out of the 196 participants, only a few (between 0 and 5) were without a total score for these questionnaires because these participants did not complete all questions within the questionnaires. The mean total scores for the Borg CR-10 Scale, HADS, and SGRQ were 4.2 (n=195), 4.9 (n=191), and 28.2 (n=196), respectively, see Table 4. Furthermore, the mean subscale scores for the HADS were 3.1 for anxiety (n=195) and 1.8 for depression (n=191). The majority of participants did not have anxiety or depression, only 11 participants experienced minor anxiety, while four participants had moderate anxiety [43]. Similarly, four participants had minor depression, and another four participants experienced moderate depression [43]. The SGRQ total and subscale scores were relatively low, indicating fewer limitations [47].” |
|
4. Did you look into association with mortality and D12 scores?
|
Thank you for this suggestion. As this is a secondary analysis of baseline data obtained from a randomized controlled trial, we did not collect information on mortality. Therefore, it is not feasible to conduct an association analysis involving mortality. |
|
5. Did you look into inhalers and d12 scores? Were these patients on inhalers, if yes how many and was their an association?
|
We did not collect any information about inhalers. |
|
6. How did the D12 scores compare with other co-existing diagnosis like Pulmonary hypertension, asthma, ILD, heart failure etc. |
Thank you for your comments. In terms of comorbidities, only 4 out of 196 participants had asthma, 6 had pulmonary tuberculosis, and 12 had heart disease. Therefore, it is not feasible to conduct comparison analyses using these comorbidities.
|

Reviewer 2 Report
Comments and Suggestions for Authors
The manuscript is very well written, and I have only two minor suggestions:
- The methodology section could benefit from including details on how the sample size was calculated for the study.
- Please provide a more detailed justification for using the HADS scale in validating the dyspnea assessment.
Author Response
|
Reviewers |
Authors’ Responses |
|
The methodology section could benefit from including details on how the sample size was calculated for the study.
|
Thank you for these comments. We have inserted the following sentence in the “Methods” section to address this point. Page 3: “The sample size estimation was determined based on the primary outcome, mBorg score, along with a 25% attrition rate observed in the pilot study [37]; the required sample size was 196, with 98 in each group.”
|
|
Please provide a more detailed justification for using the HADS scale in validating the dyspnea assessment. |
We have inserted the following sentences in the “Discussion” section to address this point. Page 12: “The convergent validity of the simplified Chinese version of the D-12 was excellent. As hypothesised, the D-12 total scores moderately or strongly correlated with the Borg CR-10 Scale scores, the NRS scores, the HADS scores, and the SGRQ scores. The D-12 includes seven physical function items and five emotional function items, which measure very similar or the same concepts as the Borg CR-10 Scale [11], the NRS (perceived severity of breathlessness) [13], the HADS [43] and SGRQ [46]. For example, both the HADS and D-12 contain emotional content for measuring psychological distress. HADS and D-12 were used to explore the association between them [26], and the HADS is commonly utilized and has been validated in cancer patients [44, 45].” |

Reviewer 3 Report
Comments and Suggestions for Authors
Adjusted title
Abstract - they could have just put a contextualisation phase on dyspnoea related to lung cancer, but I don't think that's obligatory either. As this is an article validating a scale, you could start the abstract with the objective.
The abstract is objective, it contains the aims, methodology and results obtained.
Key words -
Perhaps replace simplified Chinese with simplified Chinese Version
Perhaps include validation;
Introduction -
Objective
On line 57/58 they should put the acronym SGRQ
Take a simple approach to the different assessment instruments that exist to assess dyspnoea; describe the instrument being validated in a simple and objective way.
Conclude with the aim of the work.
Methodology
They refer to ethical issues.
The translation and linguistic validation process complies with the recommendations.
Identify and briefly characterise the different assessment instruments used in the RCT and their operationalisation.
The instrument has good Cronbach's Alpha values, indicating good internal consistency and reliability. It also has good convergent validity.
They identify the normality of the sample and the different statistical tests used.
They characterise the sample and the statistically significant relationships between the questionnaire being validated and the others used.
They clearly identify limitations and the limitations mentioned are relevant.
Conclusion - simple and objective
There are 60 appropriate bibliographical references, 13 of which date from 20202 or later and 13 from 2025 or later.
They present 2 dated between 1980 and 1990 and 6 dated between 1990 and 2000, which seems rather old to me and not always essential.
Author Response
|
Reviewers |
Authors’ Responses |
|
Abstract - they could have just put a contextualisation phase on dyspnoea related to lung cancer, but I don't think that's obligatory either. As this is an article validating a scale, you could start the abstract with the objective. The abstract is objective, it contains the aims, methodology and results obtained.
|
Thanks for these comments. We have revised and inserted the following sentences within the “Abstract” section. Page 1: “The simplified Chinese version of the Dyspnea-12 Questionnaire (D-12) has not yet been translated and validated for patients with primary lung cancer. This study aimed to evaluate the psychometric properties of the simplified Chinese version of the D-12 for patients with primary lung cancer.” |
|
Key words - Perhaps replace simplified Chinese with simplified Chinese Version Perhaps include validation; |
Thanks for this comment. Page 1: “Keywords: Dyspnea-12 Questionnaire; Psychometric Properties; Simplified Chinese Version; Primary Lung Cancer; Validation” |
|
Introduction - Objective On line 57/58 they should put the acronym SGRQ Take a simple approach to the different assessment instruments that exist to assess dyspnoea; describe the instrument being validated in a simple and objective way. Conclude with the aim of the work. |
Thanks for these comments. We have revised the following sentence: Page 2: “….the Cancer Dyspnoea Scale (CDS) [15, 16], the Chronic Respiratory Questionnaire (CRQ) [17] and Saint George’s Respiratory Questionnaire (SGRQ) [18]. ”
|
|
Methodology They refer to ethical issues. The translation and linguistic validation process complies with the recommendations. Identify and briefly characterise the different assessment instruments used in the RCT and their operationalisation. The instrument has good Cronbach's Alpha values, indicating good internal consistency and reliability. It also has good convergent validity. They identify the normality of the sample and the different statistical tests used. They characterise the sample and the statistically significant relationships between the questionnaire being validated and the others used. They clearly identify limitations and the limitations mentioned are relevant. Conclusion - simple and objective |
Thanks for these comments. |
|
There are 60 appropriate bibliographical references, 13 of which date from 20202 or later and 13 from 2025 or later. They present 2 dated between 1980 and 1990 and 6 dated between 1990 and 2000, which seems rather old to me and not always essential.
|
Thanks for this suggestion. We have incorporated more updated references into the manuscript, for example: Page 14-17: “7. Yang Y, Qian X, Tang X, Shen C, Zhou Y, Pan X, Li Y: The links between symptom burden, illness perception, psychological resilience, social support, coping modes, and cancer-related worry in Chinese early-stage lung cancer patients after surgery: a cross-sectional study. BMC psychology 2024, 12(1):463. 8. Lucas-Ruano D, Sanchez-Gomez C, Rihuete-Galve MI, Garcia-Martin A, Fonseca-Sanchez E, Fernández-Rodríguez EJ: Descriptive Study on the Relationship between Dyspnea, Physical Performance, and Functionality in Oncology Patients. In: Healthcare: 2024: MDPI; 2024: 1675. 64. Tu J, Shen M, Li Z: When cultural values meets professional values: a qualitative study of chinese nurses’ attitudes and experiences concerning death. BMC palliative care 2022, 21(1):181.” |
